# Biochemical Composition and Biological Activities of Various Population of *Brassica tournefortii* Growing Wild in Tunisia

**DOI:** 10.3390/plants11233393

**Published:** 2022-12-06

**Authors:** Hajer Tlili, Abdelkarim Ben Arfa, Abdelbasset Boubakri, Najjaa Hanen, Mohamed Neffati, Enrico Doria

**Affiliations:** 1Laboratory of Pastoral Ecosystems and Valorization of Spontaeous Plants and Microorganisms, Institute of Arid Regions (IRA), Medenine 4119, Tunisia; 2Department of Biology and Biotechnology, University of Pavia, 27100 Pavia, Italy

**Keywords:** *Brassica tournefortii*, carotenoids, antiproliferative activity, alkaloids, LC-MS, PCA

## Abstract

*Brassica tournefortii Gouan*, commonly known (Aslooz) in Tunisia, is an annual plant, native to the North Africa and Middle East. *Brassica* species are used as food, their young leaves can be cooked, providing nutrients and health-giving phytochemicals such as phenolic compounds, polyphenols and carotenoids. Phytochemical composition and bioactivity of *Brassica tournefortii* leaf extracts, collected from four different bioclimatic zones in Tunisia, are investigated in the present study. Results showed that location and climatic variations can alter the phytochemical composition of *B. tournefortii*. Interestingly, HPLC analysis enabled identifying lutein and beta-carotene at high concentrations, especially in extracts of *B. tournefortii* collected from Gabes (B2) (344 µg/g of lutein) and *B. tournefortii* collected from Zarzis (B3) (1364 µg/g of beta-carotene). In particular, the antioxidant activity measured by DPPH assay showed that the extract of the plants collected from the growing region of Zarzis exhibits the highest antioxidant activities (0.99 mg/mL). All the *Brassica tournefortii* extracts showed a relevant antiproliferative activity, especially toward the Caco-2 cell line. These preliminary data resulted in being useful to correlate growth environmental conditions with different accumulation of metabolites in Brassica species still being poorly studied.

## 1. Introduction

Brassicaceae, formally named Cruciferae, are plants including a high number of species, approximately 3709 species distributed throughout the world [1], predominantly concentrated in districts with high temperatures and achieving a maximum of diversity around the Mediterranean area [2]. Therefore, their relevant nutritional and economic value is due to their wide worldwide distribution and abundant consumption as food. Brassica buds, flowers, leaves, roots, seeds, and stems are all edible and have multiple uses separately, which make them the most widely consumed plant, yielding by-products such as fresh vegetables, oil, condiments, fodder, and preserved food, among others [3]. These species are rich in phytochemical compounds recognized as bioactive secondary metabolites (glucosinolates, phenolic compounds, flavonoids and carotenoids), as well as their richness of minerals and vitamins [4]. Various research on biological activity of carotenoids has improved the knowledge of the role of these ubiquitous pigments, which seem to play a protective role against cancer, cardiovascular diseases, cataracts, and, most of all, age-related macular degeneration [5,6]. These bioactive compounds originated from plants have manifold beneficial effects on human health, including the anti-inflammatory and antioxidant properties [7]. Carotenoids found in Brassicaceae species are well known for their positive implication in human nutrition, since they have several health benefits, including antioxidant, anti-inflammatory, anti-microbial, anti-allergic, cytotoxic, and anti-tumor activities [8].

*Brassica tournefortii*, often recognized as “Sahara mustard”, is an annual dicotyledonous herb native to arid and semi-arid regions of North Africa, Mediterranean areas of southern Europe, and the Middle-East [9]. It has been introduced to Australia and North America in the beginning of twentieth century, and it is presently regarded as a highly invasive species threatening the native annual flora [10,11]. *Brassica tournefortii* is especially common in areas with wind-blown sediments as reported by Minnich and Sanders [12]. In the same way, it is also pervading exotic annual grassland and coastal sage scrub. This plant often forms almost pure stands on abandoned sandy fields. Few surveys converging on *B. tournefortii* have elucidated that this species, such as many other Brassica species, could be used for many medicinal, culinary, and ecological intents [13].

The biosynthesis and accumulation of bioactive compounds derived from both primary and secondary metabolism can be very different among populations growing under different environmental conditions [14,15]. In order to contrast changes in some environmental factors such as the temperature, the availability of water, the amount of light radiation, the concentration of CO_2_, the soil nutrients, as well as the attack of herbivores and pathogens, plants react by the varying the content of secondary metabolites, which allow them to adapt better to the changing environmental conditions [16,17]. Many species of Brassica such as *Brassica tournefortii* were generally considered invasive weeds in many regions worldwide and remained poorly studied until recently. Chemical composition and biological activities of this plant have been little explored. Among the few studies present in literature, Curto et al. [18] showed that *B. tournefortii* seeds present a biofumigation effect against parasitic nematodes and fungi, mainly due to the presence of isothiocyanates and some flavonoids; Rahmani et al. [19] reported that the dry leaves of this species exhibit a significant beneficial effect against breast cancer. Thus, further studies are needed to provide information about the abundance, diversity, and distribution of secondary compounds of *B. tournefortii* related to environmental conditions. In addition, further studies are needed to know if there are differences in the qualitative and quantitative composition of secondary compounds in this Brassica species, due to different growth habitats. The aim of the present study was, therefore, to determine and compare the quantitative and qualitative biochemical composition and biological activities of different *B. tournefortii* populations, still poorly studied, growing in different bioclimatic areas of Tunisia: (B1) *B. tournefortii* collected from Sfax; (B2) *B. tournefortii* collected from Gabes; (B3) *B. tournefortii* collected from Zarzis; and (B4) *B. tournefortii* collected from Tatouine, highlighting how the different accumulation of some metabolites, responsible for different biological activity, is related to different environmental conditions.

## 2. Results

### 2.1. Secondary Metabolites Composition and Antioxidant Properties

The total polyphenols content (TPC), total flavonoids (TFC), total condensed tannins (TCT), carotenoids, saponins, and alkaloids in all the plant samples of different populations of *B. tournefortii*, collected in different localities in Tunisia, are presented in Table 1, while the content of carotenoids is presented in Table 2. TPC was significantly variable among the examined populations. It ranged from 9.91 ± 0.08 to 11.34 mg/g GAE DW^−1^ and from 17.78 to 21.78 mg/g GAE DW^−1^ for methanolic and acetonic extracts, respectively. As shown in Table 1, for all the different origins of Brassica populations, acetone extracts exhibited higher phenolic content, 1.8 fold higher compared to the methanolic ones. However, *B. tournefortii* collected from Zarzis (B3) showed the highest content.

Contextually, the TFC value of *B. tournefortii* B3 was significantly higher compared to the values registered for the other three populations derived from different localities. TFC ranged from 2.36 to 6.59 mg/g CE DW^−1^ for the acetone extracts, and it resulted in being around 4.55 ± 0.57 to 5.95 ± 0.21 mg/g CE DW^−1^ for the methanolic ones in all the Brassica populations. The amount of total condensed tannins (TCT) in Brassica leaf extracts was higher in *B. tournefortii* collected in Tatouine (B4) compared to those observed in plants collected in other regions B1, B2, and B3.

The content of saponins and alkaloids in leaves of *B. tournefortii* is displayed in Table 1. Collected data showed relevant levels both of accumulated saponins (especially) and alkaloids. On the other side, the content of both types of compounds was significantly different among the observed populations; in particular, *B. tournefortii* B3, collected from the middle arid region, presented the highest concentration both of saponins (5.6%) and alkaloids as well (1.8%), whereas *B. tournefortii* B1 accumulated the lowest concentration of alkaloids (0.7%), showing, however, a relevant amount of saponins (4.8%). In plants, generally, glycosylation of triterpenes, which leads to saponin biosynthesis, is a consequence of the adopted defense strategies; these response mechanisms influence physicochemical properties and biological activities of these metabolites [20], responsible for several functional properties, including anti-inflammatory, antifungal, antibacterial, antiparasitic, anticancer, and antiviral activities [21]. For this reason, these data contribute to improving the knowledge about the phytochemical composition of this plant, since there are few data available in literature about the total content of alkaloids.

Regarding carotenoid content, all extracts from different *B. tournefortii* populations accumulate significant amounts of β-carotene and lutein. In particular, the B3 population exhibited a high concentration of β-carotene, while B2 showed the highest level of lutein. *B. tournefortii* grown in Sfax (B1) showed no accumulation of beta-carotene and, contextually, the lowest concentration of lutein. The lutein and β-carotene are the dominant carotenoids in mature leaves under normal conditions that occupy nearly 50% and 25% of the total foliar carotenoids [22]. Lutein and β-carotene are generally known to be the dominant carotenoids in Brassica [23].

Recognition of many health benefits provided by phenolic compounds has encouraged scientific research to increase the knowledge about the antioxidant capacity of various plant-derived products. However, a standardized method for the determination of the antioxidant properties has not yet been established. The present study used the DPPH assay to estimate the antioxidant capacity. To our knowledge, there is no study which correlates the total antioxidant activities, due to the accumulation of metabolites in leaves, with the climatic region where the *B. tournefortii* plants were grown. The ANOVA test revealed significant differences in reducing power among the different regions. Leaves from B3 tend to be more active than those from the other growing regions across the entire studied period. It can therefore be concluded that the different antiradical power of plant leaf extracts depends both on the growth area and on the used extraction methods. In particular, acetone extraction resulted in being more efficient, compared to that one performed using methanol, in order to extract metabolites with antiradical power.

### 2.2. Anti-Inflammatory Activity

The in vitro anti-inflammatory activity of the plant extracts was assayed measuring the inhibition of thermally induced protein denaturation. Several anti-inflammatory drugs have shown dose dependent ability to inhibit thermally induced protein denaturation. As part of the investigation about the anti-inflammatory activity, the results of the ability of the examined plant extracts to inhibit protein denaturation have been studied, and results are presented in Figure 1. A significative difference was observed among the examined plant extracts; a concentration of 100 μg/mL of plant extract resulted in being very efficient in inhibiting heat-induced denaturation of albumin in an interval between 44.15 ± 0.32% and 48.05 ± 0.24%. Acetylsalicylic acid (ASA), used as a control reference, showed 46.10 ± 0.26% inhibition.

### 2.3. Antiproliferative Activity of Brassica Tournefortii Extracts

The cytotoxicity (measured as antiproliferative effect) of *B. tournefortii* leaf extracts, collected from the different four growth areas, was carried out against two human cancer cell lines, K-562 (myelogenous leukemia) and Caco-2 (colon carcinoma). Results showed in Table 3, expressed as IC_50_, revealed how the cytotoxic impact of the examined extracts could be dependent on the location, which determines a different accumulation of some anti-proliferative compounds; the cell viability percentage of the two cancer cell lines is shown in Figure 2. Out of the four examined extracts of *B. tournefortii* collected in different areas, plants collected in the Zarzis region (B3) were found to be the most efficacious in inhibiting cell proliferation (IC_50_ value = 25 µg/mL), especially for the Caco-2 colon tumor cell line. All the plant extracts showed the highest cytotoxic effects on the treated cancer cells at the concentration of 2 mg/mL. All the *B. tournefortii* extracts, at the same concentration, induced cell death in Caco-2 cell culture in a range from 52.24 (for plants collected from Sfax region, B1) to 74% (for plants collected from Zarzis region, B3), respectively. On the other hand, K-562 cell death resulted in a range varying from 55.68 (for *B. tournefortii* collected from the region of Zarziz, B3) to 58.66% (for *B. tournefortii* collected from the region of Gabes, B2).

In general, therefore, all tested extracts showed the most effective cytotoxic activity against the Caco-2 cell line, with respect to the K-562 cell line: the IC_50_ value ranged from 25 µg/mL to 140.18 µg/mL for *B. tournefortii* B3 and B1, respectively. Statistically, it was observed how the cytotoxic effect of the different extracts is strongly dependent on the growth region (on the environmental conditions) and, definitely, on the cell type.

### 2.4. Principal Component Analysis (PCA) and Clustering Analysis

Biochemical composition and biological activities’ measurements of *B. tournefortii* populations collected from different geographical localities (representing different environmental conditions) have been investigated using PCA. Depending on this analysis, the axes of inertia had been withheld, as seen in Table 4. The structuring of accessions showed 91.64% of the total variation (Figure 3). The PCA loading plot shows, at the same time, how well the principal components correlate with the original variables, and the correlations between the different activities and the different bioactive components. The first principal component (PC 1) is strongly correlated with six of the original variables: polyphenols, flavonoids, β-carotene, lutein, alkaloids contents, and anticancer activity against Caco-2 with loading of 0.831, 0.983, 0.942, 0.667, 0.672, and 0.857, respectively. Moreover, the first principal component correlates most clearly with the flavonoids and β-carotene content (correlation level of 0.983 and 0.942). The second axis reflected an anti-inflammatory, antiproliferative activity against K-562, DPPH radical scavenging activity, and saponins content with loading of 0.698, 0.776, 0.972, and 0.838 (Table 4). Figure 3 showed a good correlation between DPPH radical scavenging activity, anti-inflammatory activity, and the different bioactive components: polyphenols, flavonoids, β-carotene, lutein, and alkaloids contents; this analysis confirms that the two activities are strictly correlated, chiefly through the measured bioactive components. Oval forms exhibited in Figure 3a clustered the different populations of *B. tournefortii* (four growing regions: B1: Sfax; B2: Gabes; B3: Zarzis; B4: Tatouine) into two classes. When cluster analysis is applied, a resulting discriminate structure is evident. The different populations of *B. tournefortii* resulted in being grouped into two main clusters (Figure 3b). The lofty correlation coefficient obtained from PCA analysis proposes that the geographical distribution is related with the accumulation of secondary metabolites among populations. Thus, the population located toward southern latitudes (B2, B3 and B4) showed a higher level of phenolics compared to the populations located in the northeast area (B1). Results presented in this work are in accordance with that of Del Valle et al. [24], who also found a similar trend for *Silene littorea* populations.

## 3. Discussion

Generally, Brassica vegetables are recognized by their obvious role in human nutrition due to their phytochemicals, such as minerals, vitamins, glucosinolates, and phenolics [25]. Indeed, the Brassica sp. are high in phytochemical compounds, especially bioactive secondary metabolites, with the capacity to alter different molecular targets in the cells; in particular, Brassica plants are rich in glucosinolates (GLSs) and their breakdown products (isothiocyanates (ITCs) and indoles), together with phenolic compounds, tocopherols, and carotenoids; among others, terpenes and substantial seed oils are responsible for several important biological activities [26,27]. 

Several studies have inquired into the phenolic composition of varieties of the Brassicaceae family, but few data on phenolic composition of *B. tournefortii* are available in literature. It is well known that the synthesis, diversity, and quantity of Brassica’s phytochemicals depends on the interaction of several factors such as germination, the environmental conditions, or the nutrient supply during the crop cultivation [28,29]. The TPC found in the four analyzed plant samples was higher than that reported for two species of *Brassica oleracea*, (6.37 mM GAE/g and 2.24 mM GAE/g of Torzella and Broccolo, respectively) [30], and also higher than the values reported for seven Brassicaceae species present in Italy (4.30 to 13.80 mg of gallic acid equivalents/gof sample) [31]. The variation of phenolic contents among the plant populations may represent an adaptation mechanism to survive under variable and difficult environmental conditions, since these phytochemicals are reported to play several roles in defense against biotic and abiotic environmental stresses [32]. The biochemical variances observed among the studied populations may be the result of the different environmental conditions under which each population grows; in addition, it has been noticed that environmental factors affect the expression of genes involved in the synthesis of phytochemicals [33]. The evident correlation coefficient resulting from PCA analysis proposes that the geographical distribution is correlated with the concentration of some secondary metabolites. Hence, the populations located toward southern latitudes (B2, B3 and B4) showed a higher level of phenolics compared to the populations located in the northeast area (B1). Results presented in this work are in agreement with those of Del Valle et al. [24], who also found a similar trend for *Silene littorea* populations. 

The climatic conditions referred to precipitation level and temperature and were different in each collection site (Table 5). High temperatures were recorded in the area B4, whereas the lowest was in B1 and B2. Some previously published reports demonstrated how temperature and soil features are major factors that regulate the synthesis of phenolic compounds [34,35,36]. TFC level is higher in the plant samples collected from Zarzis (B3), although similar to those collected in Gabes (B2) and Tataouine (B4). Finally, B1 populations presented the low amount of TFC.

Tannins are natural polyphenols that possess an increased antioxidant activity than other phenols. Plants grown in B1 and B2 areas presented a similar level of TCT, whereas their values were different from those found in *B. tournefortii* B3 and B4; these data suggest that the environmental context has less impact on the accumulation of foliar flavonoids and tannins, compared to the type of solvent used during the extraction process, highlighting the difference in polar fraction of the extracts.

The TFC determined in the current report was higher than that found in *Pituranthos tortuosus* (Coss.) Maire (2.92 mg RE/g DW), an edible plant widely consumed in some semi-arid areas of Tunisia [37]. Flavonoids, the best-known group of phenolic compounds, are generally found in all plant parts, and it is well known how their human intake is related to health benefits [38]. These compounds perform key roles in plants, such as protectors against solar UV radiation, pathogens, and herbivores [39]. Then, TCT levels measured in the leaves of the four studied *B. tournefortii* populations were lower than those reported by Benabderahim et al. [37], who quantified higher content of these compounds in *Ephedra alata* (7.81 mg TAE/g DW), another edible plant typical of arid regions in Tunisia. As mentioned, there is not much data available in literature about the content of saponins and alkaloids in *B. tournefortii*; for this reason, a comparison of the obtained results with those registered in literature is not possible.

Regarding the level of carotenoids, results of HPLC analysis are presented in Table 2. Overall, two carotenoids were identified and quantified in all extracts at high concentrations: lutein and β-carotene. Generally, these two compounds are reputed to be the dominant carotenoids in Brassica species [40]. On the other side, according to the literature, there is not much available data about the content of saponins and alkaloids in *B. tournefortii*; for this reason, a comparison of the obtained results with those registered in literature is not possible.

As previously mentioned, many factors such as genotype, environment, growth stage and time of harvest, processing and storage conditions, and method of analysis can affect the phenolic profiles of plant tissues [41]. It is also known that geographical factors such as altitude and latitude influence the composition of defensive chemicals including phenolic compound levels [42]. Recent studies have shown a monthly variation in polyphenol concentrations with the highest levels occurring in summer at noon when drought, temperature, and UV radiation are highest, suggesting that flavonoids have a role in the protection against UV rays and water stress in Mediterranean endemic species [43,44]. Accumulation of natural antioxidant components and the consequent biological activity of plant extracts highly depend on type and concentration of the extracting solvent [45]. In fact, brassica vegetables have powerful antioxidant properties assigned to the high levels of carotenoids, vitamins, and especially phenolic compounds [46]. 

Regarding the cytotoxic activity of *B. tournefortii* extracts, a relevant activity against Caco-2 and K-562 cell cultures was observed for all the plant extracts, confirming the data recorded by Moustafa et al. [47] for two Brassicaceae species: *Brassica nigra* and *Matthiola Arabica*. However, it is important to underline how the results reported by Rahmani et al. [48] showed weak and no cytotoxic activity against human colon cancer, breast, and ovarian cancer of *B. tournefortii* extracts, obtained from different organs (leaves, stems, and roots). Then, in this work, it is well shown that the examined extracts, rich in carotenoids, were found to be more effective in inhibiting the growth of colon cancer cells compared to one of leukemic cells (Figure 1). Moreover, a relevant variation in the IC_50_ values of B2, B3, and B4 extracts, in particular, against the two different cell lines was observed. Previous studies assigned this selectivity to the sensitivity of the cell lines to the nature of the active compounds present in these extracts or to the tissue specific reaction [49,50].

In some paper, lutein showed a cell growth inhibitory and cytotoxic effects in several cancer cell lines and animal models [51,52]. It was demonstrated that lutein inhibited the growth of rat prostate carcinoma cells (AT3 cells) and human prostate cancer cells (PC3) [53], it induced apoptosis in transformed but not in normal human mammary cells [54], and, moreover, it altered mouse mammary tumor development [55]. Previous studies showed as well that lutein has relevant anti-inflammatory, antioxidant, and anti-cancer properties [56]. There is not so much published data regarding the antiproliferative activity of lutein, generally considered as a safe (GRAS) and non-toxic phytochemical agent, against colon cancer cells [57]. Anti-proliferative properties of carotenoids may be considered as the result of combination of their antioxidant properties with their ability to interact with specific regulators of cell signaling pathways [7]. Carotenoids play a vital role in maintaining the human wellness. For this reason, they are considered essential dietary nutrients. In addition to the extreme environmental conditions possibly altering the accumulation and the status of many nutritionally important metabolites, including carotenoids in plants [58], the level of foliar carotenoids may vary in response to some environmental conditions such as light, temperature, and drought stresses, typical of arid regions where the plant samples were collected.

Data presented in this work confirm a higher accumulation of carotenoids in plant populations grown in the southernmost regions of Tunisia, probably due to the need to protect the foliar apparatus against the high solar radiation. At the same time, the high temperature and important level of water stress determined a higher accumulation of alkaloids in plant populations collected in more arid regions. In general, the limitation in soil water content produces prominent impacts on plant architecture and physiology, also including alteration of secondary metabolites’ accumulation such as carotenoids. This later performs a vital role in drought stress signaling, neutralizing oxidative stress, and acclimation in plants such as carotenoid-derived phytohormone abscisic acid induces stomatal closure, inhibiting transpiration whenever plants receive drought and salinity signals [59]. Therefore, the low level of precipitations in the Zarzis region allows a high accumulation of carotenoids and phenolic components, especially in the areal part of plants, determining a potent antiproliferative activity of the extracts. It was observed that one of the major environmental parameters influencing the secondary metabolism in plants (accumulation and typology of phytochemicals) is the fluctuation in air temperature [60,61]; at room temperature, corresponding to the temperature of the surrounding environment, the level of accumulated carotenoids is species and tissue specific. In conclusion, preliminary data collected in this work showed how the climatic condition of the regions where B2, B3, and B4 Brassica grew determined a general accumulation of secondary metabolites, responsible for a more evident biological activity.

Obtained results from this study represent an input to deepen the knowledge about the variety and profusion of phytochemicals of wild plant growing in arid zones and on the environmental factors responsible for the variations of biochemical profile of a given plant species. Collecting data that relates the environmental factors with the plant biochemical composition may contribute to finding strategies to improve and increase the accumulation of specific secondary metabolites in plants, especially in species of food and medicinal interest, as Rahmani et al. [48] pointed out. 

## 4. Materials and Methods

### 4.1. Plant Materials

The plant material (*B. tournefortii*) was provided by the Arid Lands Institute of Medenine, Tunisia (IRA), where it was harvested in April 2017 from four growing regions chosen along a transect of increasing aridity: Sfax (B1) (upper arid bioclimate), Gabes (B2) (middle arid), Zarzis (B3) (middle arid), and Tatouine (B4) (lower arid zone), located in southeast Tunisia. The areas where the plants were collected, according to the distribution of *B. tournefortii* species reported by Rahmani et al. [48], are located along the coast of the Gulf of Gabes and in some southwestern areas of Tunisia. Young leaves were collected from these locations in April 2017, after the rainy season, when the leaves of this species appear. The species of the leaf samples was confirmed by Mohamed Neffati (Laboratory of Pastoral Ecosystems and Valorization of Spontaneous Plants and Microorganisms, Medenine, Tunisia), and voucher specimens are deposited at the herbarium of the IRA. 

*Environmental conditions of collection sites*. In order to compare the environmental conditions affecting the synthesis and accumulation of secondary metabolites of *B. tournefortii*, data about the level of the annual average precipitation (Prec), referred to a period of three years (2015–2017), and the maximum and minimum temperature values (Tmx and Tm, respectively) were collected. Table 5 showed the data about the geographical coordinates and weather conditions for the sampling locations.

### 4.2. Sample Preparation

The collected leaf samples were grounded at around 100 mesh, using a Retsch S/S Cross Beater Hammer Mill Sk1, then dried in the shade at RT (25–28 °C) for two weeks and finally stored in dark conditions until use. Methanol 70% and acetone 70%, which have different characteristics of polarity, were used to obtain two plant extract solutions (1 g/10 mL), in order to compare the different content of metabolites; the plant material was then macerated for 24 h in shaking conditions (50 rpm, 40–45 °C) and used to quantify the total content of polyphenols, the total content of flavonoids, and the total antioxidant activity (DPPH test). For other biological assays, condensed tannins, carotenoids, saponins, and alkaloids analysis, the extraction method is described in each section. 

### 4.3. Chemical Characterization

#### 4.3.1. Total Phenolic Content

The total content of phenolic (TPC) was measured according to the method described by Medoua et al. [62]. For each sample, methanol and acetone extracts (pH 2.5 using HCl) were centrifuged at 6000 rpm for 10 min; the supernatants were collected, and the residue pellets were further washed with 1.5 mL of acetone 70% employing mechanical agitation (800 rpm, 30 min at 4 °C) and then centrifuged. The resulting supernatants were assayed using the Folin–Ciocalteu reagent. Absorbance was measured at 725 nm, and results were expressed in Gallic Acid Equivalents (GAE) using a gallic acid standard curve.

#### 4.3.2. Total Flavonoid Content

The total flavonoids content (TFC) was estimated using the Lola-Luze method as described by Lola-Luze et al. [63], slightly modified as described below. An aliquot of 150 µL of the extract was blended with 600 µL of water and 45 µL of sodium nitrite NaNO_2_ (5%). After 10 min of incubation, 45 µL of 10% aluminium trichloride (AlCl_3_) in methanol was added, incubated for 2 min, and then 300 µL of NaOH (1 M) with the same volume of water were added. After 10 min, samples were spectrophotometrically assayed at 510 nm (PerkinElmer UV–VIS spectrophotometer) against a blank sample consisting of an extract solution with 1395 mL of methanol without AlCl_3_. Total flavonoid content is expressed as mg of catechin equivalents/g of dry material. 

#### 4.3.3. Condensed Tannins Content

Condensed tannins content (TCT) was quantified according to the method used by Doria et al. [64]. An amount of plant powder (0.5 g) was mixed with 10 mL of acetone/methanol (containing 1% HCl) solution (7:3) and shaken (800 rpm) for 1 h in the dark at 60 °C. The samples were then sonicated and centrifuged at 6000 rpm for 10 min. An aliquot (0.5 mL) of each extract was mixed with 3 mL of butanol: HCl (95:5, *v*/*v*) solution in screw-capped test tubes and incubated at 95 °C for 60 min. The absorbance was then read at 550 nm. All results were expressed as mg of standard delphinidin equivalents/g dry material. 

#### 4.3.4. Carotenoids Content

Carotenoids content was determined by the method as described by Kurilich [65] with modifications performed by the authors of the present work and described below. Dry plant material (0.1 g) was added to 25 mL of a chloroform: ethanol: diethyl ether solution (2:1:0.5) containing BHT. Potassium hydroxide KOH (1 mL, 80% *w*/*v*) was supplemented to the mixture for saponification, and the samples were wiggled for 1 h. Then, the solution was transferred to a separator funnel, on which 30 mL of chloroform: ethanol (2:1) solution was added. After layer separation, the upper layer was washed with 50 mL of 5% NaCl and subsequently dried by a rotavapor. The residue was then resuspended with n-hexane and chromatographically assayed. 

#### 4.3.5. Saponins Content

The total content of saponins was measured by the gravimetric method as depicted by Kaur et al. [66]. Twenty grams of each sample were mixed with methanol 70% (100 mL) and macerated for 24 h in dark conditions and then divided into water and n-butanol (1:1 ratio) solution. This obtained solution was moved to a separator funnel and retained for 2 h. The upper n-butanol layer was separated, and the solvent was evaporated to obtain crude saponin extract. The results are expressed as a percentage of the weight.

#### 4.3.6. Alkaloids Content

The total content of alkaloids was evaluated according to the method described by Biradar et al. [67] with some modifications showed below. An amount of 5 g of each sample was macerated in ethanol: acetic acid (45:5) solution for 4 h at a room temperature of 25 °C). After centrifugation at 4000 rpm, an organic phase (ethanol, acetic acid) is collected and subjected to evaporation using a rotavapor system. The residue (acetonic phase) is solubilized with 2 mL of 3% H_2_SO_3_ solution and with 8 mL of water. The obtained solution was poured into a funnel separator, using 10 mL of petroleum ether: diethyl ether (1:1) solution to remove lipids fraction. After separation, the lower phase was collected and then mixed with 30 mL of NH_4_OH (final pH = 8) and 30 mL of chloroform to determine the release of alkaloidal bases. The final solution was filtered in the presence of Na_2_SO_4_ and then dried with a rotavapor.

### 4.4. LC-Analysis

Chromatographic analysis (HPLC) of lutein and β-carotene was performed using a Shimadzu system, consisting of an LC-20AD XR binary pump system, SIL-20AC XR autosampler, CTO-20AC column oven, and DGU-20A 3R degasser (Shimadzu, Kyoto, Japan), equipped with a DiscoVery BIO Wide Pore C18-5 column (Thermo Electron, Dreieich, Germany, 15 cm × 4.6 mm, 5 μm) and a PDA detector (SPD-M20A). The used solvents were: (A) methanol: 1 M ammonium acetate 8:2 and (B) methanol: acetone 8:2. The injection volume was 20 µL and the flow rate 1 mL/min. UV absorbance was settled at 450 nm. The gradient for elution was linear from 0 to 100% B in 20 min; after 5 min, 100% of A was used for a further 5 min. Finally, a linear flow of 100% A for 5 min was used to equilibrate the column. The content of lutein and beta-carotene was measured and quantified using different concentrations of the standard sample of the two carotenoids (concentration range was from 0.01 to 1 mg/mL for both lutein and beta-carotene).

Standards were purchased from Fluka Analytical.

### 4.5. Biological Activities

#### 4.5.1. DPPH Test

Measure the anti-radical power of the prepared extracts by means of the widely used 2,2-Diphenyl-1-Picryl-hydrazyl (DPPH) test. Different volumes of the samples (from 25 to 75 µL) were added to 1 mL of 0.2 mM DPPH solution and to a pure methanol solution for a total volume of 1.5 mL. After 1 h in the dark, the absorbance was read at 517 nm against a methanol control, and the results were presented as EC_50_ (mg/mL) [68], which is defined to be the extract concentration required to scavenge 50% of the DPPH radical. The lower the value of EC_50_, the greater the antioxidant capacity of the sample:% Radical scavenging activity = (Control OD − Sample OD)/Control OD

#### 4.5.2. In Vitro Anti-Inflammatory Activity (Inhibition of Albumin Denaturation)

The reaction mixture consisted of 200 µL of *B. tournefortii* leaf extracts and 750 µL of 1% aqueous solution of bovine albumin [69]. The pH of the reaction mixture was adjusted using a small amount of 1 N HCl. The sample extracts were incubated at 37 °C for 20 min and then heated to 57 °C for further 20 min. After cooling, the turbidity of samples was measured spectrophotometrically at 660 nm. The experiment was performed in triplicate. Percent *inhibition* of protein denaturation was calculated using the following Formula: % *inhibition* = [{*Abs control* − *Abs sample*}/*Abs control*] × 100(1)
where *Abs control* is the absorbance without sample, and *Abs sample* is the absorbance of sample extract/standard. ASA (acetylsalycilic acid) was used as positive control. 

### 4.6. Cell Culture

The human acute monocytic leukaemia K-562 (ATCC^®^ TIB-202™) cell line was obtained from the Riken Cell Bank (Tsukuba, Ibaraki, Japan). Cells were cultured in RPMI-1640 medium (Gibco™, Catalog No. 11875093) supplemented with 10% (*v*/*v*) heat-inactivated fetal bovine serum (FBS), 1% Penicillin–Streptomycin, 1% L-glutamine (200 mM) and 0.05 mM 2-mercaptoethanol. Cells were maintained at 37 °C and 5% CO_2_ in a humid atmosphere. All experiments were performed with cells found in an exponential growth.

The human Colorectal Adenocarcinoma Caco-2 (ATCC^®^ HTB-37™) cell line was obtained from the Riken Cell Bank (Tsukuba, Ibaraki, Japan). Cells were cultured in DMEM medium (Gibco™, Catalog No. 11875093) supplemented with 10% fetal bovine serum (Sigma, St. Louis, MO, USA), 1% penicillin-streptomycin (Lonza Walkersville, Inc., Basel, Switzerland), 1% non-essential amino acids, and 1% L-Gluthamine and incubated at 37 °C, 5% CO_2_. All experiments were performed with cells found in an exponential growth.

### 4.7. Cytotoxicity Assay

In order to investigate the antiproliferative effect of the examined plant extracts towards the K-562 and Caco-2, cell viability and proliferation were assessed using 3-(4,5-dimethylthiazol-2-yl)-2,5-diphenyl-tetrazolium bromide or MTT assay [70,71]. In brief, Caco-2 cells and K-562 cells were incubated as a culture in Dulbecco’s modified Eagle’s medium in 96-well plates (2 × 10^5^ cells/well) at 37 °C with 5% of CO_2_ for 24 h. The medium was then replaced with another medium containing the extracts from each plant in the final concentration of 100 μg/mL. After 48 h incubation, MTT (5 mg/mL PBS) containing medium (0.45 mg/mL final concentration) was added. The plates were then incubated at 37 °C for 24 h. Sodium dodecyl sulphate (SDS; 10% *v*/*v*) was then added to each well (100 μL), followed by overnight incubation at 37 °C. This reagent was used to solubilise and detect the formazan-crystals, and its low concentration was determined by optical density. Absorbance was obtained at 570 nm using a microplate reader (Powerscan HT; Dainippon Pharmaceuticals USA Corporation, East Windsor, NJ, USA). Data are presented as a percentage of cell proliferation against a control (100% of cell proliferation) using different concentrations of the plant extracts. Experimental conditions are given in the Appendix A in detail. 

### 4.8. Statistical Analysis

A descriptive analysis was performed to describe the entire results within each kind of test. Concerning the antiproliferative activity, an unimpaired Student’s *t*-test was used to compare treated cells with control cells. Regarding the biochemical composition analysis, antioxidant activity, and the in vitro anti-inflammatory activity, a one-way analysis of variance (ANOVA one-way) followed by DUNCAN test was performed to test possible significant differences among mean values from different populations’ locality of *B. tournefortii*. The level of significance was set at *p* < 0.05 for all analyses. Statistical analyses were performed using SPSS v.20 software.

## 5. Conclusions

Undoubtedly, the Brassica vegetables are an excellent source of pharmaceuticals, and could be studied in vitro or in vivo to be exploited according to their phytochemical diversity as well as by the wide variety of beneficial effects on nutrition and human health. Thus, the present study revealed that a number of positive effects of African mustard, such as phytochemicals, were found, which is beneficial for one’s health. The phytochemicals such as alkaloids, flavonoids, saponins, tannin, and terpenoids were present, which increases the medicinal potential of African mustard and thus can be used for the treatment of various diseases. They may therefore have beneficial health effects and can be considered as chemopreventive or adjuvant agents against cancer. However, modern medicine may have many side effects, and many times it is resulting not completely safe for human consumption; consequently, it could be better to adopt natural remedies which generally has no side effects and are considered safe for human consumption.

## Figures and Tables

**Figure 1 plants-11-03393-f001:**
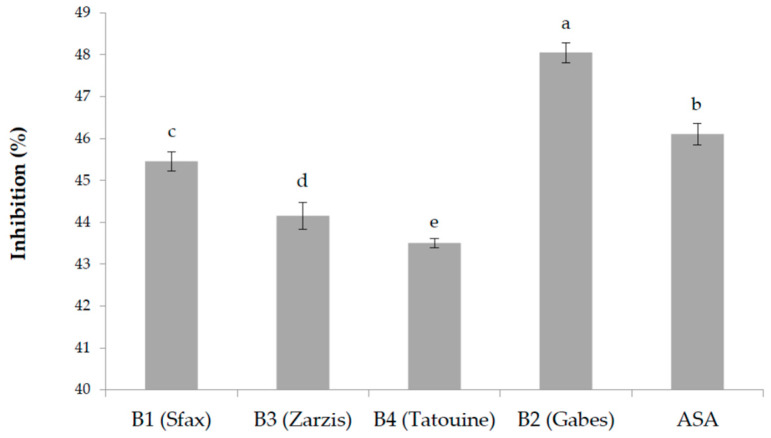
Anti-inflammatory activity of the different methanolic extracts (100 μg/mL) of the tested *B. tournefortii* plants, measured as an inhibitory effect of protein denaturation (%). Data are the mean of three replicates ± SD. Different superscript letters indicate significant differences between *B. tournefortii* growing regions at 5% according to the Duncan’s multiple range test and ANOVA test.

**Figure 2 plants-11-03393-f002:**
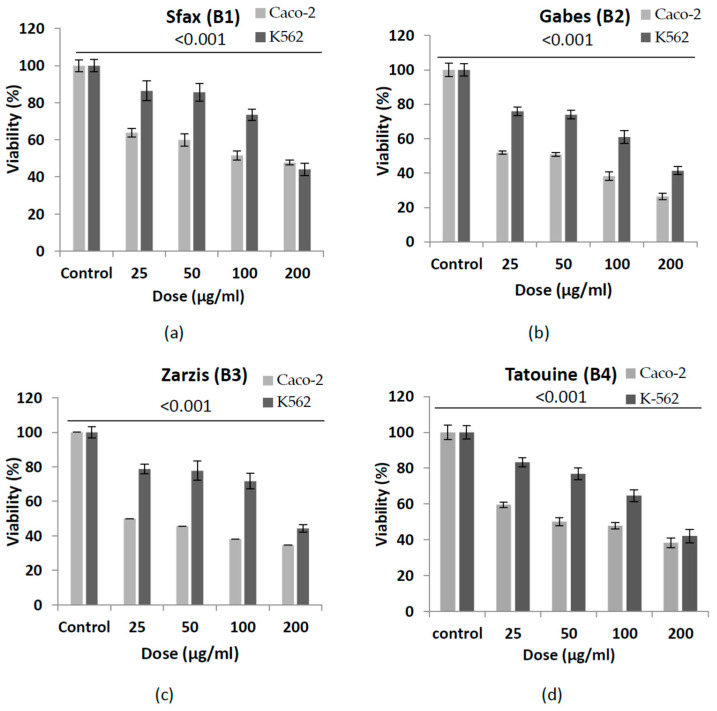
Antiproliferative activity of the four extracts *of Brassica tournefortii* growing in different regions of southern Tunisia, tested on two neoplastic cell lines (K-562 and Caco-2). (**a**) *B. tournefortii* samples collected from Sfax (B1); (**b**) *B. tournefortii* samples collected from Gabes (B2); (**c**) *B. tournefortii* samples collected from Zarzis (B3); (**d**) *B. tournefortii* samples collected from Tatouine (B4). Data are the mean of three replicates ± SD. The unpaired Student’s *t*-test indicates significant differences compared with the untreated cells (control) (*p* < 0.001).

**Figure 3 plants-11-03393-f003:**
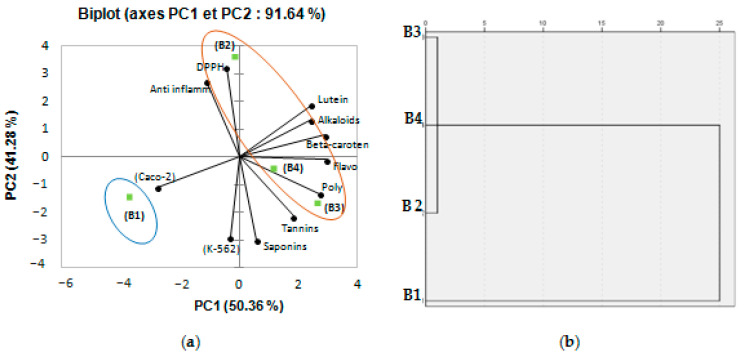
(**a**) Principal components analysis biplot (axes F1 and F2: 91,64%); (**b**) dendrogram comparing measured bioactive components and biological activities of different populations (four geographical localities of *B. tournefortii*, B1: Sfax; B2: Gabes; B3: Zarzis; B4: Tatouine).

**Table 1 plants-11-03393-t001:** Phytochemical composition of plant extracts of five wild populations of *Brassica tournefortii* leaf extracts.

Growing Region	*TPC* (mg/g)		*TFC* (mg/g)		*TCT* (mg/g)	*DPPH* (*EC* 50, mg/mL)		*Crude Saponins*%	Alkaloids%
	Methanol 70%	Acetone 70%	Methanol 70%	Acetone 70%		Methanol 70%	Acetone 70%		
**B1 (Sfax)**	9.91 ± 0.08 ^d^	17.78± 0.68 ^c^	4.55 ± 0.57 ^bc^	2.36 ± 0.046 ^c^	2.04 ± 0.07 ^b^	1.23 ± 0.062 ^bc^	1.57 ± 0.025 ^bc^	4.80 ± 0. 04 ^b^	0.7 ± 0.01 ^d^
**B2 (Gabes)**	10.52 ± 0.096 ^b^	18.44 ± 0.14 ^c^	5.33 ± 0.33 ^b^	4.69 ± 0.51 ^b^	1.96 ± 0.04 ^b^	2.45 ± 0.012 ^a^	3.76 ± 0.80 ^a^	3.20 ± 0.02 ^d^	1.7 ± 0.01 ^b^
**B3 (Zarzis)**	11.34 ± 0.012 ^a^	21.78 ± 0.98 ^a^	5.95 ± 0.21 ^a^	6.59 ± 0.35 ^a^	2.20 ± 0.03 ^a^	1.03 ± 0.072 ^d^	0.99 ± 0.085 ^d^	5.60 ± 0.03 ^a^	1.8 ± 0.04 ^a^
**B4 (Tatouine)**	10.48 ± 0.08 ^c^	20.25 ± 0.86 ^b^	5.30 ± 0. 24 ^b^	6.08 ± 0.25 ^a^	2.25 ± 0.02 ^a^	1.71 ± 0.17 ^b^	1.90 ± 0.27 ^b^	4.20 ± 0.02 ^c^	1.2 ± 0.03 ^c^

Data are the mean of three replicates ± SD. Different superscript letters above the values in the same line indicate significant differences at 5% according to the Duncan’s multiple range test and ANOVA test.

**Table 2 plants-11-03393-t002:** Carotenoids level measured by HPLC. Results of *B. tournefortii* leaf extracts are presented as µg/g of dry weight.

Compounds	RT	B1 (Sfax)	B2 (Gabes)	B3 (Zarzis)	B4 (Tatouine)
**Beta-carotene (µg/g)**	21.095	-	1070.81 ± 1.48 ^c^	1361.87 ± 3.62 ^a^	1102.34 ± 1.23 ^b^
**Lutein (µg/g)**	4.222	21.547 ± 1.04 ^d^	343.55 ± 2.16 ^a^	283.24 ± 1.75 ^b^	275.88 ± 2.59 ^c^

Data are the mean of three replicates ± SD. Different superscript letters above the values in the same line indicate significant differences at 5% according to the Duncan’s multiple range test and ANOVA test.

**Table 3 plants-11-03393-t003:** IC_50_ (µg/mL) of four *B. tournefortii* extracts tested on Caco-2 and K-562 cell lines.

	IC_50_ (µg/mL)			
	B1 (Sfax)	B2 (Gabes)	B3 (Zarzis)	B4 (Tataouine)
Caco-2	140.18 ± 2.3 ^d^	42.62 ± 2.09 ^b^	25 ± 1.14 ^a^	64.2 ± 2.02 ^c^
K-562	183.12 ± 4.13 ^c^	156.78 ± 2.77 ^a^	185.40 ± 3.80 ^c^	165.04 ± 3.27 ^b^

Data are the mean of three replicates ± SD. Different superscript letters above the values in the same line indicate significant differences at 5% according to the Duncan’s multiple range test and ANOVA test.

**Table 4 plants-11-03393-t004:** Correlations between variables and factors.

	F1	F2	F3
TPC	**0.831**	0.166	0.003
TFC	**0.983**	0.001	0.016
TCT	0.379	**0.423**	0.197
Beta-carotene	**0.942**	0.058	0.000
Lutein	**0.667**	0.330	0.002
Saponins	0.040	**0.838**	0.122
Alkaloids	**0.672**	0.168	0.160
DPPH	0.022	**0.972**	0.006
Anti-inflammatory activity	0.137	**0.698**	0.165
(Caco-2)	**0.857**	0.110	0.033
(K-562)	0.010	**0.776**	0.213

**Table 5 plants-11-03393-t005:** Sampling sites and climate conditions (annual averages, 2015–2017, for each location) for four collection regions of *Brassica tournefortii* from south Tunisia.

Collection Region	Latitude	Longitude	Altitude (m)	Prec (mm)	T Max (°C)	T Min (°C)	Climate
**Sfax (B1)**	34.7398° N	10.7600° E	23	319.05	23.59	19.30	Upper arid
**Gabes (B2)**	33.8881° N	10.0975° E	4	146.00	24.52	19.09	middle arid
**Zarzis (B3)**	33.5041° N	11.0881° E	18	127.00	24.43	20.42	middle arid
**Tataouine (B4)**	32.9211° N	10.4509° E	247	178.71	26.84	17.00	Lower arid

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
