# Peer review of "Biochemical Composition and Biological Activities of Various Population of *Brassica tournefortii* Growing Wild in Tunisia"

_plants, 2022, doi:10.3390/plants11233393_

Round 1
Reviewer 1 Report
The manuscript entitled “Biochemical composition and biological activities of various population of Brassica tournefortii growing wild in Tunisia” is devoted to a phytochemical investigation on Brassica tournefortii leaves collected from four regions with different сlimatic conditions. I have to constate the manuscript contains plenty of inaccuracies in the results discussion as well as the conlusions are not representative. There are some comments:
1. The ordinal numbers of the Tables and the order of their appearance in the text should be checked, because the first table in the text is Table 2, the next one - also Table 2 etc.
2. L80, L88: the values 8.95 GAE DW-1 of TPC content and 4.74 CE DW-1 of TFC content are absent in the Table
3. L83: "a 0.5 % more compared to" - the phrase looks strange with regard to each value.
4. L89: what do authors mean saying "The amount of leaves (TCTC) was"?
5. L90: "The levels of saponins and alkaloids in leaves are displayed in Table 3" - Table3 is absent.
6. L95: B1 population, but not B2 showed the lowest alkaloids content.
7. L137: Units of measurements of carotenoid contents should be specified in Table2
8. In the legend to Fig.1 the concentrations and type of extracts should be specifed
9. L184, 187: The terms "cytotoxic" and "antiproliferative" activity are used by authors as synonyms, but in Experimental part the conditions of MTT test (time of exposition) are not shown, that doesn't give the opportunity to discuss the antiproliferative activity.
10. Legend to Fig. 2: "Antiproliferative activity of four Brassica tournefortii populations" - populations can not demonstrate any activity.
11. L324,334,453 etc: "total condenced tanins content" are differently abbreviated (TCTC, TCT, CTC) in the text.
12. L263-272: "If the two plots (biplot) were gathering together, it is seen that the high Bioactive components measured is related to the growing regions B2, B3 and B4 and the region B1 showed the highest cytotoxic activity and the high saponins content." It seems that PCA results contradict to the other experiments: B1 showed the lowest cytotoxicity IC50 140.18 and 183.12 (Table 4)! The saponin content (4.80%) is also not highest (5.60%) (Table1).
13. L324-327: "TFC and TCT found in the plants grown in areas B1 and B2 were similar, whereas their values were different from those found in B. tournefortii B4 and B5." - B5 population is absent in all the experiments; TFC of B2 and B4 were similar, but not for B1 and B2; CTC of B1 and B2 were also not very close to each other and, in addition, didn't significantly differ from B3 and B4
Generally, the manuscript contains a lot of comparisons with diverse literature data but the own observations of the authors are not satisfactory analyzed, the conclusions about the influence of the certain environmental factors to the secondary metabolites composition of populations B1-B4 are not made. The question: "What is the optimal growth conditions for the plants to contain the maximal quantity of targeted compounds", - is not answered.
Author Response
Dear revewer 1
Thank you for your precious and careful revision that highlighted several carelessness and typos. We are sorry for that. Especially the comments at point 9, 12 and 13 resulted important to improve and correct some inaccuracies. All the observations were used to modify the manuscript according to your suggestion and are highlighted in text. We modify the methods regarding MTT test and also the discussion section was improved following your comments, in particular focusing more attention to the relationship between the growth environmental conditions and accumulation of secondary metabolites. Even if in some case the difference of concentration is not significative among some plants, anyway the group of the plants collected in southern region showed a highest biological activity compared to that one collected in Sfax area. We underline that these are preliminary data but anyway interesting to associate some environmental stress with accumulation of some metabolites involved in biological activities of a Brassica species poorly investigated but with high potential. The text was extensively revised in all sections and improved in language.

Reviewer 2 Report
Everywhere in the text:
- the name of the plant should be written in italic.
- the names of the plants should be written correctly: the names of the authors are missing.
- “mL” should be written instead of “ml”.
- “min” should be written instead of “minutes”.
- “h” should be written instead of “hour” or "hours".
- all data should by written to 0.01.
- all values are recorded with a point and not a comma.
in: Abstract
- it should be included concrete values for individual components.
- some corrections of the text are necessary, because it is not clear enough now.
in: Keywords
- some corrections of the text are necessary, because it is not clear enough now.
in: Introduction
- the aim of the study is not clear.
in: Results
- numbering of tables should start from 1.
- data about chemical composition in the investigated leaves are missing.
in: Discussion
- the explanation about different chemical composition of the extracts is missing.
in: Materials and methods
- the moisture content (%) of the leaves, the method for its determination and the conditions of the drying should be specified.
- it should be specified what “room temperature” is.
- it should be explained why these solvents for extraction of the leaves are used (methanol 70% and acetone 70%).
- it is not clear what the temperature of maceration is.
- NaNO2 should be written instead of NaNo2.
- it should be specified who made the modifications of the methods by Kurilich et al. [65].
- AlCl3 should be written instead of AlCl3.
- n-hexane should be written instead of hexane.
- it should be specified what “RT” is.
- H2SO3 should be written instead of H2SO3.
- Na2SO4 should be written instead of Na2SO4
in: Conclusions
- some corrections of the text are necessary, because it is not clear enough now.
in: References
- the list of references should be written according to the guideline for authors.
Author Response
Dear revisor
Thank you for your precious and careful revision that highlighted several carelessness and typos. We are sorry for that. We modified the paper according to your observations and comments. We modified the abstract section, making clearer the aim of the work in the introduction section too. We improved the methods sections with more information, adding the reason why we used two different solvents for metabolites extraction. The discussion and conclusion section were also improved, heavily extended and made it clearer. Alle the modifications are highlighted in the text.
Specifically:
the moisture content (%) of the leaves, the method for its determination and the conditions of the drying should be specified.
- we didn’t calculate moisture content (%). We just dried the samples, removing all the water content.
data about chemical composition in the investigated leaves are missing.
Preliminary data about the biochemical composition of the different populations are presented in table 1 and table 2.

Round 2
Reviewer 1 Report
Despite the authors have improved the quality of manuscript, there are still plenty of grammatic and stylistic mistakes that complicate understanding of the text. Some data are repeatedly provided.
So, it seems to me that extensive English editing is required.
Author Response
The manuscript entitled “Biochemical composition and biological activities of various population of Brassica tournefortii growing wild in Tunisia” is devoted to a phytochemical investigation on Brassica tournefortii leaves collected from four regions with different сlimatic conditions. I have to constate the manuscript contains plenty of inaccuracies in the results discussion as well as the conlusions are not representative. There are some comments:
Dear revisor
Thank you for your precious and careful revision that highlighted several carelessness and typos. We are sorry for that. Especially the comments at point 9, 12 and 13 resulted important to improve and correct some inaccuracies. All the observations were used to modify the manuscript according to your suggestion and are highlighted in text. We modify the methods regarding MTT test and also the discussion section was improved following your comments, in particular focusing more attention to the relationship between the growth environmental conditions and accumulation of secondary metabolites. Even if in some case the difference of concentration is not significative among some plants, anyway the group of the plants collected in southern region showed a highest biological activity compared to that one collected in Sfax area. We underline that these are preliminary data but anyway interesting to associate some environmental stress with accumulation of some metabolites involved in biological activities of a Brassica species poorly investigated but with high potential
- The ordinal numbers of the Tables and the order of their appearance in the text should be checked, because the first table in the text is Table 2, the next one - also Table 2 etc.
Thank you for the observation. It was corrected
- L80, L88: the values 8.95 GAE DW-1 of TPC content and 4.74 CE DW-1 of TFC content are absent in the Table
Corrected
- L83: "a 0.5 % more compared to" - the phrase looks strange with regard to each value.
The sentence was more properly arranged, making more clear
- L89: what do authors mean saying "The amount of leaves (TCTC) was"?
The sentence was rearranged correctly
- L90: "The levels of saponins and alkaloids in leaves are displayed in Table 3" - Table3 is absent.
Sorry for mistake, everything was properly corrected
- L95: B1 population, but not B2 showed the lowest alkaloids content.
Yes, correct. Thank you. Text is properly modified.
- L137: Units of measurements of carotenoid contents should be specified in Table2
Yes, thank you. Done
- In the legend to Fig.1 the concentrations and type of extracts should be specified
Yes, correct. It was properly modified
- L184, 187: The terms "cytotoxic" and "antiproliferative" activity are used by authors as synonyms, but in Experimental part the conditions of MTT test (time of exposition) are not shown, that doesn't give the opportunity to discuss the antiproliferative activity.
We modified the section according to your suggestion. We used antiproliferative activity to specify the biological property, however it is correlated to the cytotoxicity of extract. MTT was described in the section (from line 579)
- Legend to Fig. 2: "Antiproliferative activity of four Brassica tournefortii populations" - populations can not demonstrate any activity.
Correct. Thank you. Modified
- L324,334,453 etc: "total condenced tanins content" are differently abbreviated (TCTC, TCT, CTC) in the text.
It was uniformed in TCT
- L263-272: "If the two plots (biplot) were gathering together, it is seen that the high Bioactive components measured is related to the growing regions B2, B3 and B4 and the region B1 showed the highest cytotoxic activity and the high saponins content." It seems that PCA results contradict to the other experiments: B1 showed the lowest cytotoxicity IC50 140.18 and 183.12 (Table 4)! The saponin content (4.80%) is also not highest (5.60%) (Table1).
These data were not presented very well. Text was properly modified to make more clear this part. The biuological effects are related not to only one class of compounds probably but is the result of interactions. Anyway some data are clearlu showing how some climatic condition such as low rain or high solar radiation determinate accumulation of specific compounds which are related to some biological activity. The discussion section was extensively modified and commented
- L324-327: "TFC and TCT found in the plants grown in areas B1 and B2 were similar, whereas their values were different from those found in B. tournefortii B4 and B5." - B5 population is absent in all the experiments; TFC of B2 and B4 were similar, but not for B1 and B2; CTC of B1 and B2 were also not very close to each other and, in addition, didn't significantly differ from B3 and B4
Sorry for this typos. The text was modified and all the carelessness properly corrected.
Generally, the manuscript contains a lot of comparisons with diverse literature data but the own observations of the authors are not satisfactory analyzed, the conclusions about the influence of the certain environmental factors to the secondary metabolites composition of populations B1-B4 are not made. The question: "What is the optimal growth conditions for the plants to contain the maximal quantity of targeted compounds", - is not answered.
Thank you for this analysis. Comparison with literarure were useful to highlight how poor is literature about data of some brassica species and, when data are present, it is interesting doing comparisons which in many case are helpful to correlate compounds with some biological activity. Anyway, the discussion section was extensively modified. In this part It was mentioned that one of the major environmental parameters influencing molecular and biochemical phenomena including the secondary metabolism in plants (accumulation and typology of phytochemicals), is the fluctuation in air temperature. at room temperature, corresponding to the temperature of the surrounding environment, the level of accumulated carotenoids is species and tissue specific. In conclusion, preliminary data collected in this work showed how the climatic condition of the regions where B2 B3 and B4 Brassica grew determined a general accumulation of secondary metabolites responsible of a more evident biological activity. Definitely, these are preliminary data requiring more investigation, but represent a first important observation about some correlation about climatic conditions and biochemical profile of different Brassica populations.

Reviewer 2 Report
in: Abstract
- it should be included concrete values for individual components.
- the names of the plants should be written correctly: the names of the authors are missing. It should be written Brassica tournefortii Gouan.
in: Results
- all data should by written to 0.01.
- data about chemical composition in the investigated leaves are missing.
- mL should be written instead of ml.
in: Materials and methods
- the moisture content (%) of the leaves, the method for its determination and the conditions of the drying should be specified.
- it should be specified what “RT” is – room temperature??
- it should be specified what “room temperature” is.
- it should be specified who made the modifications of the methods by Biradar et al. [67].
- the compounds H2SO3 and AlCl3 should be written correctly.
Author Response
Comments and Suggestions for Authors
Everywhere in the text:
Dear revisor
Thank you for your precious and careful revision that highlighted several carelessness and typos. We are sorry for that. We modified the paper according to your observations and comments. We modified the abstract section, making clearer the aim of the work in the introduction section too. We improved the methods sections with more information, adding the reason why we used two different solvents for metabolites extraction. The discussion and conclusion section were also improved, heavily extended and made it clearer. Alle the modifications are highlighted in the text.
- the name of the plant should be written in italic.
done
- the names of the plants should be written correctly: the names of the authors are missing.
It was corrected
- “mL” should be written instead of “ml”.
done
- “min” should be written instead of “minutes”.
done
- “h” should be written instead of “hour” or "hours".
done
- all data should by written to 0.01.
done
- all values are recorded with a point and not a comma.
done
in: Abstract
- it should be included concrete values for individual components.
done
- some corrections of the text are necessary, because it is not clear enough now.
The text was extensively modified, in all the sections.
in: Keywords
- some corrections of the text are necessary, because it is not clear enough now.
Thank you for the comment, I agree that some modifications were necessary to make more clear the text. Extensive modifications were made in all the section, especially in discussion and results, in order to make more clear some parts. You can find all modifications highlighted in the text.
in: Introduction
- the aim of the study is not clear.
Agree. Text modified.
in: Results
- numbering of tables should start from 1.
Sorry for typos. Everything was corrected
- data about chemical composition in the investigated leaves are missing.
Preliminary data about the biochemical composition of the different populations are presented in table 1 and table 2.
in: Discussion
- the explanation about different chemical composition of the extracts is missing.
The discussion section was extensively modified and improved. It was mentioned that one of the major environmental parameters influencing molecular and biochemical phenomena including the secondary metabolism in plants (accumulation and typology of phytochemicals), is the fluctuation in air temperature. at room temperature, corresponding to the temperature of the surrounding environment, the level of accumulated carotenoids is species and tissue specific. In conclusion, preliminary data collected in this work showed how the climatic condition of the regions where B2 B3 and B4 Brassica grew determined a general accumulation of secondary metabolites responsible of a more evident biological activity. Definitely, these are preliminary data requiring more investigation, but represent a first important observation about some correlation about climatic conditions and biochemical profile of different Brassica populations.
in: Materials and methods
- the moisture content (%) of the leaves, the method for its determination and the conditions of the drying should be specified.
- We didn’t calculate Moisture content (%). We just dried the samples, removing all the water content. Anyway the section was improved according to your suggestions. (from line 474)
- it should be specified what “room temperature” is.
We specified in the section that room temperature is the temperature of the surrounding environment
- it should be explained why these solvents for extraction of the leaves are used (methanol 70% and acetone 70%).
Correct observation. We specify in the text that 2 extractions solvents are useful to compare the different concentration of compounds can be extracted from the material and, moreover, to test the different polarity of the extracted compounds. In facts using two different solvents we noticed a different concentration of some compounds. In particular, as mentioned in the text, acetone extracts resulted to be more efficient, compared to the methanol, to extract metabolites with antiradical power.
- it is not clear what the temperature of maceration is.
- The tempreture of maceration is 40-45°C
- NaNO2 should be written instead of NaNo2.
done
- it should be specified who made the modifications of the methods by Kurilich et al. [65].
Done. We modified the method described in that manuscript
- AlCl3 should be written instead of AlCl3.
Ok, done
- n-hexane should be written instead of hexane.
Ok, done
- it should be specified what “RT” is.
Ok, done
- H2SO3 should be written instead of H2SO3.
Ok, done
- Na2SO4 should be written instead of Na2SO4
Ok, done
in: Conclusions
- some corrections of the text are necessary, because it is not clear enough now.
Agree. We extensively modified text, making it more clear and improving all the sections, the language and the data and comment explanations. You can see all the highlighted modifications in the text
in: References
- the list of references should be written according to the guideline for authors.
Done
